# Learning Population-Level Representations with Joint Embedding Predictive Architectures

## Abstract

Multivariate population data is ubiquitous across scientific and real-world domains, arising in settings where the identity of a system is revealed through the composition of its constituent samples. For example, a patient's clinical state can be inferred from the joint analysis of their blood cells, while the properties of a galaxy can be characterized from the distribution of its stars and their spectra. To our knowledge, attempts to learn representations of such data remain limited, largely because its inductive structure is subtle, making feature extraction particularly challenging. Inspired by recent advances in joint embedding predictive architectures, we challenge the prevailing assumption that population-level data lacks sufficient signal for representation learning, and show that by leveraging both the compositional structure of the data and the properties of individual samples, rich and expressive representations can indeed be learned. We demonstrate our approach in the biomedical domain, addressing the long-standing challenge of scaling machine learning to large single-cell transcriptomics datasets for patient representation.

## 1 Introduction

Self-supervised learning (SSL) (Schmidhuber, 1990; Bengio et al., 2013; Hadsell et al., 2006; Grill et al., 2020) leverages structural properties of data to learn meaningful and transferable representations without manual labels. Its most notable successes have been in computer vision (Grill et al., 2020; Caron et al., 2021; Oquab et al., 2024; Chen et al., 2020), where the image domain is equipped with a rich set of symmetries and local structure: semantic concepts are spatially grounded, and invariances such as translation and rotation can be directly exploited (Bronstein et al., 2021). These structural properties provide strong inductive biases, making SSL objectives particularly effective.

### 1.1 Representing Multivariate Population Data

Extending SSL beyond vision is more challenging, as many domains lack such explicit geometric structure. This is especially true for unordered data, such as sets, where permutation invariance and compositionality rather than spatial invariance is the primary structural property (Zaheer et al., 2018). Yet sets and other data types built on them-appear frequently in practice. One important case arises when each datapoint is itself a set of elements, and the identity of the datapoint is revealed through the composition of those elements.

We refer to this setting as multivariate population data. It consists of a base collection of objects, each characterized by a distribution over multivariate random variables. In other words, every object is represented not by a single vector but by a population of vectors. Examples include users in recommendation systems, defined by the set of their transactions, and patients in biomedical data, who could be characterized by the collection of their cells molecular profiles (Kulkarni et al., 2019; Ianevski et al., 2024). Such data is abundant and important, yet its subtle structure makes it a particularly challenging domain for representation learning.

Though this data lacks local geometric properties, it is characterized by three key structural aspects: permutation invariance, compositionality, and the statistical regularities of the underlying

distributions. We posit that the latter two provide sufficiently strong priors to support effective representation learning. To harness them, we turn to joint embedding architectures, and in particular, joint embedding predictive architectures (JEPA) (Assran et al., 2023; 2025; Huang et al., 2025). Originally developed in vision, JEPA learns embeddings of a context that are predictive of withheld content—for example, encoding part of an image so that it is informative about the representations of masked patches. Unlike generative methods, which aim to reconstruct masked tokens pixel by pixel, JEPA predicts representations, encouraging the network to capture higher-level concepts while discarding fine-grained details. **We argue that this predictive objective can be naturally adapted to multivariate population data, where the goal is to learn embeddings of subsets that are informative about the rest of the population.**

## 1.2 BIOMEDICAL PROOF OF CONCEPT: SC-JEPA

To put these ideas to the test, we develop SC-JEPA, a joint-embedding predictive architecture for biomedical data. We focus on single-cell transcriptomics, which enables large-scale measurement of gene expression in individual cells of the human body. In this setting, each patient is represented by the distribution of their cells and molecular properties, and the task is to use this information to infer a meaningful patient-level embedding. Our approach does so by partitioning each patient's data into a context subset and a feature subset, learning a representation of the context, and training the model to predict the representation of the feature subset from it. This predictive objective encourages the embedding to capture population-level structure that goes beyond simple aggregation, while remaining robust to permutation and technical variation.

In the context of patient representation, we seek embedding spaces that satisfy three key desiderata.

**(1) Integration across datasets.** We want embeddings that bring together data from multiple sources in a way that reflects biological rather than technical variation. This is particularly important because individual datasets typically include too few patients to support robust training. A foundational model must therefore integrate across cohorts into a common space that is minimally influenced by batch effects while preserving genuine biological distinctions.

**(2) Patient stratification.** We want embeddings that amplify inter-patient differences so that clinically meaningful heterogeneity can be uncovered. This is crucial for precision medicine: many treatment errors stem from an incomplete understanding of how immune responses vary across individuals. Embeddings that sharpen these differences enable stratification of patients into groups that may respond differently to therapy.

**(3) Predictive and transferable representations.** Finally, we want embeddings that are predictive of future or unseen conditions and transferable across contexts. To probe this, we analyze drug-response data across cancer cell lines, asking whether embeddings trained on some lines can predict the effect of a drug on a withheld line. Success here suggests that embeddings capture system-level properties rather than dataset-specific patterns.

Taken together, these desiderata define the properties of a useful embedding space for patient-level representation learning. Our results show that predictive objectives over subset representations can satisfy these goals. In particular, our approach, SC-JEPA, enables robust dataset integration, more precise patient stratification, and accurate prediction of counterfactuals, providing a foundation for applications that span from personalized medicine to drug discovery.

## 2 RELATED WORK

### 2.1 MULTIVARIATE POPULATION DATA

We define *multivariate population data* as a data structure in which each datapoint corresponds not to a single observation but to a *set* (or population) of multivariate samples. Formally, an object $o_i$ is represented by a collection $S_i = \{x_{i1}, \ldots, x_{in}\}$, where each element $x_{ij} \in \mathbb{R}^d$ is drawn from an underlying distribution associated with that object. Examples include patients described by their sets of single-cell molecular profiles, users characterized by their transaction histories, or molecules represented by collections of local descriptors.

A natural connection arises to the field of multi-instance learning (MIL), originally introduced by Dieterich et al. (1997) in the context of drug activity prediction. In MIL, the input is a "bag" of instances with an associated label, and the objective is to classify bags rather than individual instances. Since then, numerous MIL methods have been developed, including kernel-based approaches (Gärtner et al., 2002), attention mechanisms (Ilse et al., 2018), and graph formulations (Zhou & Xu, 2007). However, MIL has largely remained a supervised classification paradigm, with methods focused on mapping bags to task-specific labels. This framing does not directly address the problem of unsupervised representation learning, where the goal is to capture population-level structure in a transferable embedding space.

To ensure permutation invariance, most approaches for set-structured data employ pooling strategies such as mean, sum, or max aggregation (Zaheer et al., 2018). Pooling is computationally efficient and often strong in practice, explaining the competitiveness of simple baselines in domains. However, pooling has intrinsic limitations, as it tends to average out fine-grained structure and suppress high-frequency variation that may carry important signal. This motivates the need for alternative approaches that move beyond pooling to capture richer signals in multivariate population data.

## 2.2 Joint Embedding Architectures

One of the central paradigms of self-supervised learning (SSL) is the joint embedding architecture (JEA), where multiple views of an input are mapped into a shared space and trained to align. Prominent methods such as SimCLR (Chen et al., 2020) , BYOL (Grill et al., 2020), and DINO (Caron et al., 2021; Oquab et al., 2024) operate by matching representations of augmented views of the same object—for example, a global view and a cropped or masked partial view. These approaches have proven highly effective, particularly in vision, where carefully designed augmentations (cropping, color jitter, patch masking) reliably preserve semantic content while altering superficial details.

However, their success depends critically on the quality of the augmentations. If the partial view is too small or uninformative, the model may learn to ignore the broader context. This reliance on carefully curated augmentations poses challenges for domains where natural augmentations are hard to define, such as unordered set-structured data.

An alternative is the joint embedding predictive architecture (JEPA) (Assran et al., 2023), which trains context embeddings to predict the representations of withheld subsets rather than directly aligning paired views. This formulation relaxes the dependence on augmentations, since subsets can be generated by natural partitioning of the data. Moreover, the predictor of JEPA readily accommodates auxiliary information into the input of the predictor module, which can be clinical metadata, and typically enjoys faster training. These advantages motivate our investigation of JEPA in the setting of multivariate population data.

## 3 Methodology

To test our assumptions about how to train self-supervised models on multivariate population data, we implemented SC-JEPA, a joint embedding predictive architecture applied to single-cell transcriptomics. While we focus on the biomedical domain, most components of the architecture are general and can be applied to any dataset where each datapoint is represented as a set of multivariate samples.

**JEPA objective.** The central idea behind JEPA is to learn representations of data that are informative about its subsets. Given an object $o_i$ represented by a set of samples $S_i = \{x_{i1}, \ldots, x_{in}\}$, we partition it into two disjoint subsets: a **context** $C_i$ and a **target** $T_i$. An encoder maps each subset to a summary representation:

$$s_C = \text{encoder}(C_i), \quad s_T = \text{encoder}(T_i).$$

A predictor then takes the context representation together with pointer tokens $\pi_T$ identifying the target subset, and outputs a predicted target summary:

$$\hat{s}_T = \text{predictor}(s_C, \pi_T).$$

The model is trained so that the predicted target matches the the true target embedding, using an $L^1$ loss:

$$\mathcal{L}_{\text{JEPA}} = \left\| \hat{s}_T - \text{sg}(s_T) \right\|_1,$$

where $\text{sg}(\cdot)$ denotes the stop-gradient operator.

**Context and target subsets.** To create context and target subsets, at each training iteration we sample a minibatch of elements from $S_i$ and randomly split them in a 20:80 ratio into target ($T_i$) and context ($C_i$) subsets. Both subsets are passed through architecturally identical encoders, which map each element through a two-layer MLP followed by an eight-layer Transformer. The encoder outputs a set of token embeddings as well as a pooled summary representation. To stabilize training, we adopt a teacher-student setup: the context encoder is updated via gradient descent, while the target encoder is updated as an exponential moving average (EMA) of the context encoder's weights, and its outputs are treated as stop-gradient targets.

**Predictor.** The predictor network conditions on both the context and the special *pointer* tokens. We concatenate the context tokens with the pointer tokens and feed them into a four-layer Transformer, which outputs a prediction of the target embedding. At this stage we do not apply any pooling operators on the context or target embeddings; thus, they preserve the same cardinality as in the original split.

An additional advantage of this architecture is that it naturally accommodates *auxiliary sources of information*, such as metadata. We leverage this property by introducing dedicated *dataset tokens*, which are added to the context embeddings. The role of these tokens is to absorb dataset-specific variation, thereby freeing the context embeddings to represent the underlying biological signal.

**Pointer tokens.** Unlike images, sets lack spatial coordinates, so we cannot "point" to a masked region with positional encodings. To address this, we create *pointer tokens* $\pi_T$ that softly identify the target subset $T_i$ without revealing its full content. These tokens are constructed through the following procedure recently suggested by (Bizeul et al., 2025) for vision:

1. Fit PCA within the minibatch;
2. Zero out principal components until 70% of the variance is removed for elements in $T_i$;
3. Project the data back into the original feature space;
4. Pass the result through an MLP to obtain pointer embeddings $\pi_T$.

This process retains general information about the identity of $T_i$ which is enough to specify, which subset the predictor should target, while discarding most fine-grained content, preventing trivial copying. By construction, the pointer tokens are permutation-invariant and serve as an index for the predictor.

We note that alternative augmentation strategies could also be applied. For example, in single-cell expression data one could use binomial subsampling or dropout-style zeroing of features. However, the PCA-based strategy proposed here is more general.

**Variance regularization.** In addition to the JEPA loss, we apply variance regularization following the VICReg framework (Bardes et al., 2022). This term enforces that the dimensions of the learned embeddings maintain non-trivial variance across the batch, thereby preventing collapse to a degenerate representation. Concretely, in our implementation, given a batch of embeddings $\{z_k\}$, the variance penalty is defined as

$$\mathcal{L}_{\text{var}} = \frac{1}{d} \sum_{j=1}^{d} \max\left(0, \gamma - \text{Var}(z_{:,j})\right),$$

where $d$ is the embedding dimension, $z_{:,j}$ denotes the $j$-th dimension across the batch, and $\gamma > 0$ is a variance target. Our final objective combines the JEPA prediction loss and this variance term.

**Population embedding** At inference time, we compute a single embedding for each population by mean-aggregating the token embeddings produced by the context encoder. This *population embedding* serves as the final representation of the object (e.g., a patient).

## 4 EXPERIMENTS

We systematically evaluated the properties of the embeddings learned with SC-JEPA across several single-cell datasets. Our primary focus was to compare predictive architectures such as JEPA against both non-predictive joint embedding architectures (JEA) and simple aggregation-based baselines.

As a representative JEA, we implemented a DINOv2-style approach (Oquab et al., 2024) adapted to this benchmark. This model consists of teacher and student encoders with the same architecture as the SC-JEPA context encoder. Training uses two student views—a *global* view (80% of the cells) and a *local* view (20% of the cells)—mirroring the 20:80 splits used in SC-JEPA.

For aggregation-based baselines, we considered (i) vectors of cell-type proportions (aggregation of curated features), (ii) the mean of the top 50 PCA components per sample, and (iii) the mean of VAE embeddings (scVI (Lopez et al., 2018)) per sample. These baselines capture standard strategies for summarizing populations via cell type summarization or averaging of principal components or VAE embeddings.

| | Batch integration | | Biological conservation | | | | | |
|---|---|---|---|---|---|---|---|---|
| Method | Silhouette ↓ | ARI ($r{=}0.1$) ↓ | COVID–19 Recall ↑ | COVID–19 Prec. ↑ | SLE Recall ↑ | SLE Prec. ↑ | CT MSE ↓ | CT $R^2$ ↑ |
| JEPA | **0.031** | **0.389** | 0.914 | 0.890 | 0.936 | **1.000** | 1.388 | 0.699 |
| DINOv2-like | 0.094 | 0.497 | **0.934** | **0.940** | **0.968** | 0.909 | **1.277** | **0.723** |
| Pooled-PCA | 0.156 | 0.609 | 0.874 | 0.917 | 0.903 | 0.966 | 1.421 | 0.692 |
| Pooled-VAE (scVI) | 0.033 | 0.416 | 0.854 | 0.860 | 0.839 | 0.867 | 3.082 | 0.332 |
| Cell-type proportions | 0.053 | 0.692 | 0.934 | 0.825 | 0.968 | 0.682 | – | – |

Table 1: Comparison of embedding methods across two embedding properties. **Batch integration** is assessed using the silhouette score and ARI (lower is better). **Biological conservation** is assessed via (i) disease classification performance (recall and precision for COVID-19 and SLE), and (ii) cell-type proportion prediction (mean squared error, lower is better; $R^2$, higher is better).

### 4.1 EMBEDDING DIVERSE BLOOD DATASETS USING SC-JEPA

We first evaluated how well SC-JEPA embeds patient-level PBMC samples into a shared representation space. Such embeddings must satisfy two requirements: (i) *technical integration*—minimal dependence on dataset-specific technical variation; and (ii) *biological conservation*—retention of meaningful patient-level biological signal. To construct a comprehensive benchmark, we aggregated all PBMC datasets from the *cellxgene census* (CZI Cell Science Program et al., 2025) together with the *Sound Life cohort* from the Allen Institute atlas (Gong et al., 2024).

**Technical integration.** We quantified batch mixing using two standard metrics: the *silhouette score* computed on dataset labels (lower is better), and the *adjusted Rand index (ARI)* between patient clusters (Leiden resolution $r{=}0.1$) and dataset labels (lower is better). As shown in Table 1, SC-JEPA achieves the strongest overall integration, closely followed by VAE (scVI) embeddings, while PCA performs worst. Representative embedding spaces for SC-JEPA, DINOv2-like, and PCA are shown in Fig. 1, illustrating that SC-JEPA produces well-mixed technical batches.

**Biological conservation.** Because aggressive batch correction can suppress genuine biological signal, we next tested whether embeddings support discrimination between disease and healthy states. We focused on COVID-19 and systemic lupus erythematosus (SLE), training linear SVMs on patient embeddings and reporting precision/recall. For COVID-19, the DINOv2-like JEA baseline achieved the highest scores, with SC-JEPA and PCA close behind. For SLE, however, SC-JEPA performed best, with PCA ranking second. These results highlight the inherent trade-off between integration and conservation, and show that predictive self-supervision retains disease-relevant information while reducing batch effects more effectively than simple aggregation.

**Cell type prediction.** To further assess biological conservation, we asked whether embeddings preserve fine-grained compositional structure. We performed a linear probing task in which embeddings were to predict CLR-transformed cell type proportions for each patient (Aitchison, 1986; Van Den Boogaart & Tolosana-Delgado, 2013). Performance was evaluated using mean squared

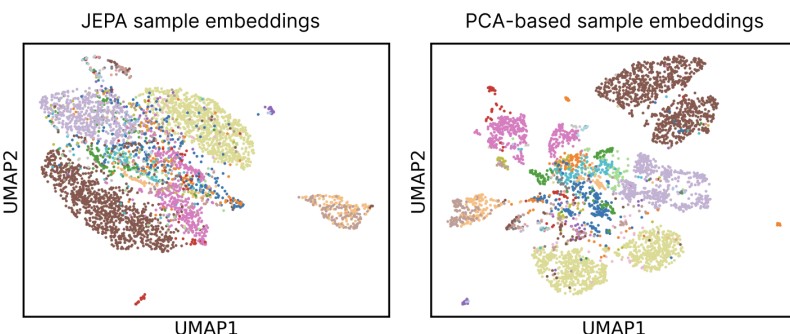

Figure 1: Patient-specific cytokine responses in the SC-JEPA space.

error (MSE; lower is better) and $R^2$ (higher is better). As shown in Table 1, the DINOv2-like approach achieved the strongest results, closely followed by SC-JEPA and PCA, while pooled VAE embeddings lagged significantly.

Across PBMC datasets, both JEA and SC-JEPA deliver the most balanced performance: they achieve top-tier batch integration while maintaining strong disease-level separability, and consistently surpass aggregation-based baselines. Between the two, SC-JEPA provides markedly stronger batch integration, while the DINOv2-like JEA shows a slight advantage on some biological readouts.

## 4.2 PATIENT STRATIFICATION INFORMED BY SC-JEPA EMBEDDINGS

We next applied SC-JEPA to a large-scale dataset of peripheral blood mononuclear cells (PBMCs) from the *10 Million Human PBMCs* collection provided by Parse Biosciences (Parse Biosciences). This dataset comprises blood samples from 12 patients, each exposed to a panel of cytokine stimulations. Such datasets are particularly important for personalized medicine, as they capture how individuals differ in their immune responses to specific activating molecules. These individualized response profiles have direct implications for suggesting effective treatments and for understanding the mechanisms that underlie immune-mediated disease.

Developing embedding methods that faithfully capture such patient-specific variation is therefore a critical step. If embeddings can reflect these individualized properties, they provide a foundation for stratifying patients according to their immune responses, ultimately supporting both treatment personalization and mechanistic discovery in biomedicine.

| Method | Batch integration | | Biological conservation | | Patient stratification | | |
|---|---|---|---|---|---|---|---|
| | Silhouette ↓ | ARI ($r$=0.1) ↓ | CT MSE ↓ | CT $R^2$ ↑ | CV Mean ↑ | CV Max ↑ | AUC–Sil ↑ |
| JEPA | **−0.015** | **0.336** | 0.101 | 0.805 | **0.886** | **1.933** | **7.057 ± 0.319** |
| DINOv2-like | 0.306 | 0.688 | 0.095 | 0.810 | 0.408 | 0.915 | 2.529 ± 0.133 |
| Pooled-PCA | 0.314 | 0.598 | **0.090** | **0.821** | 0.447 | 0.912 | 2.971 ± 0.107 |
| Pooled-VAE (scVI) | 0.300 | 0.539 | 0.188 | 0.622 | 0.457 | 1.047 | 3.550 ± 0.129 |
| Cell-type proportions | 0.531 | 0.978 | – | – | 0.499 | 1.199 | 3.932 ± 0.165 |

Table 2: Comparison of embedding methods across three representation properties. Batch integration is assessed using the silhouette score and ARI at $r$=0.1 (lower is better). Biological conservation is assessed via cell-type proportion regression (mean squared error, lower is better; $R^2$, higher is better). Patient stratification is assessed using the coefficient of variation (CV) of perturbation distances across donors (higher is better) and the AUC of silhouette scores from $k$-means clustering (higher is better).

**Batch integration and biological conservation.** As a first step, we analyzed the *10 Million Human PBMCs* dataset using the same evaluation strategy as in the previous section. Specifically, we measured batch integration with the silhouette score and ARI on donor labels, and assessed biological conservation through disease-state discrimination and cell type prediction. This allowed us to

directly compare the performance of SC-JEPA against JEA and aggregation baselines on a dataset with distinct sources of variation arising from cytokine stimulation.

The results showed that SC-JEPA performed substantially better than all other methods in terms of batch integration, achieving a near-zero silhouette score and an ARI more than 1.5 times smaller than the next best method. In terms of biological conservation, SC-JEPA trailed slightly behind PCA and DINOv2-like, with an $R^2$ of 0.805 compared to 0.821 for the best-performing baseline. Overall, these findings indicate that SC-JEPA delivers a marked improvement in batch integration—effectively removing dataset-specific variation from the latent space—while still preserving the majority of biologically relevant signal.

**Patient stratification.** We next turned to the central goal of this experiment: assessing how well the embeddings capture differences in donor-specific perturbation responses. For each donor, we computed a matrix of pairwise distances between embeddings of cytokine perturbations. We then quantified how variable these distances are across donors using the coefficient of variation (CV), reporting both the mean and the maximum CV across cytokine pairs. To assess whether these differences reflect systematic immune response patterns, we further applied $k$-means clustering (with $k = 2$–20) to each donor's embeddings and evaluated cluster quality using the silhouette score. We summarized discriminability by computing the AUC of silhouette scores across cluster numbers.

This analysis revealed that SC-JEPA achieved substantially higher CV values than competing methods, as well as higher silhouette AUC, demonstrating that its embeddings more effectively capture donor-specific differences in cytokine response. In other words, SC-JEPA produces a latent space that is not only well-integrated across donors but also finely resolves individualized response patterns.

Figure 2 illustrates this effect by showing donor-to-donor variability in cytokine co-variation within the learned space. We highlight four representative stimuli (IL-2, BAFF, M-CSF, GM-CSF; a subset of the 92 assayed), computing their pairwise distances for two donors. In Donor A, (M-CSF, GM-CSF) cluster with BAFF, suggesting association between macrophage and B cell-stimulation responses. In Donor B, the same cytokines instead cluster with IL-2, suggesting a macrophage–T cell axis. Clinically, this implies that in some patients macrophage activation is coordinated with B cell stimulation, while in others it is coordinated with T cell stimulation. Similar patterns can also be observed with embeddings based on cell-type proportions, though with greater noise and weaker clustering (Supplementary Fig. S1).

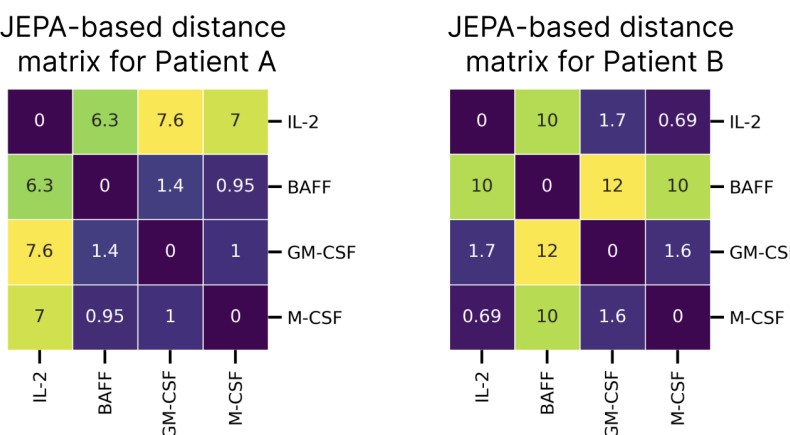

Figure 2: Patient-specific cytokine responses in the SC-JEPA space.

### 4.3 PREDICTING DRUG RESPONSE OF UNSEEN CELL LINES IN THE SC-JEPA EMBEDDING SPACE

Finally, we asked whether the latent space learned by SC-JEPA can support predictive tasks—specifically, whether it can be used to forecast the future state of a system under perturbation. To test this, we turned to the Tahoe collection (Zhang et al., 2025), a large-scale compendium of over 100 million single-cell profiles spanning 50 cancer cell lines subjected to 380 perturbations, each measured at three concentrations.

We designed a leave-one-cell-line-out experiment: one cell line was withheld during training, and the model was trained on the remaining lines. For the held-out line, we then trained a lightweight MLP to predict the embedding of a perturbed state from the embedding of its unperturbed baseline. This setup probes whether the embedding space contains enough transferable structure to generalize drug responses to unseen cellular contexts.

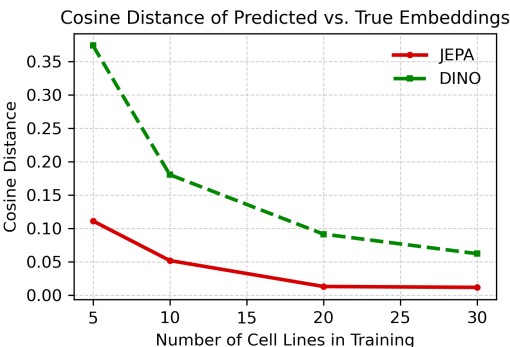

Figure 3: Prediction of drug responses in unseen cell lines. Cosine distance between predicted and true embeddings is reported for SC-JEPA and a DINOv2-like approach.

To assess the role of dataset scale, we trained models with progressively larger subsets of the available cell lines, ranging from 5 up to 30. Results ( Fig. 3) show that SC-JEPA produces embeddings that enable more accurate prediction of drug responses than the non-predictive JEA approach, as measured by cosine distance between predicted and true embeddings. Moreover, performance improved consistently as more training cell lines were included, highlighting that both SC-JEPA and JEA benefit from scale in both data diversity and predictive accuracy.

## 5 DISCUSSION

In this work we introduced SC-JEPA, a joint-embedding predictive architecture for learning representations of multivariate population data, with a focus on single-cell transcriptomics. Our central hypothesis was that permutation invariance, compositionality and feature distribution provide a sufficient inductive structure to enable predictive self-supervision.

Across diverse benchmarks, our results support this claim: SC-JEPA learns an embedding space that preserves biologically meaningful variation while being minimally influenced by technical artifacts. Strong integration is particularly critical for the decentralized future of biomedicine (Lähnemann et al., 2020; The Tabula Sapiens Consortium, 2022), where data will be generated worldwide and models must operate robustly across heterogeneous sources (McMahan et al., 2023).

Beyond integration, we showed that SC-JEPA captures donor-specific immune differences, a prerequisite for advancing personalized medicine. By resolving individual variation in cytokine response, the model highlights clinically relevant couplings between immune pathways, suggesting a path toward embeddings that reflect the personalized state of a patient's immune system.

Finally, we demonstrated that the learned embedding space supports predictive tasks, including drug-response generalization to unseen cell lines. This opens the door to rethinking drug development in terms of optimal control theory (Kirk, 1970) in representation space: rather than training

senerative models to simulate molecular profiles, one could instead identify drug combinations that steer patient embeddings toward target regions associated with desired outcomes.

## LLM Usage Disclosure

Large language models (LLMs) were used in the preparation of this manuscript. Specifically, LLM assistance (OpenAI ChatGPT, GPT-5, 2025 (OpenAI, 2025)) was employed for (i) editing and polishing text for clarity and readability, (ii) suggesting alternative phrasings to improve flow, and (iii) helping restructure sections for coherence. All technical content, experimental design, implementation, analyses, and conclusions were conceived and carried out by the authors. The authors take full responsibility for the correctness and originality of the work.

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

# A    Supplementary

Figure S1: Patient-specific cytokine responses based on differences in cell type proportions.

