# OpenReview forum: "Learning Population-Level Representations with Joint Embedding Predictive Architectures"
_ICLR.cc/2026/Conference — Submitted to ICLR 2026_

### Official Review · Reviewer_JKCY · 2025-10-25

**Soundness:** 2
**Presentation:** 2
**Contribution:** 2
**Rating:** 2
**Confidence:** 5

**Summary:**

The authors present scJEPA, a representation model for patient representation in single-cell transcriptomics. Patients are modeled as bags of cells, where the underlying task is to retrieve set-embeddings reflecting clinical and biological variation while retaining as little technical confounding. To derive patient embeddings, the authors use the JEPA representation paradigm, where cellular representations within a population are predicted from a context embedding learnt with a context encoder. The authors showcase the model on multiple datasets, evaluating the effectiveness of JEPA-like embedding strategies and embedding qualities through batch effect removal, biological conservation, and perturbation prediction metrics.

**Strengths:**

I find the scientific question quite relevant, and I commend the authors for looking into it. With the increase in the size and clinical relevance of patient-based transcriptomics datasets, learning population-level representations is a very compelling task that research should converge on. Moreover, I believe that the choice of using JEPA is well-justified based on its properties in relation to the characteristics of the data at hand. Finally, the authors use large and cutting-edge datasets, making their effort more significant.

**Weaknesses:**

Unfortunately, there are some aspects of the paper that lead me to a negative assessment. I am more than happy to discuss the criticism with the authors during rebuttals, as I may miss some points or insights that could steer my judgment upward.

1. **General opinion.** In general, I feel that the paper does not propose a strong methodological contribution. The core idea is to evaluate how JEPA translates to a new field. The main methodological and task-specific modification I could detect was the use of PCA for the pointer embedding. I am usually not against publishing simple model adaptations to machine learning conferences, but I think such a type of work should be compensated with very convincing experimental evidence, and I could not find it here. I will elaborate more on my feedback on the experiments below.

2. **Introduction structure.** In my opinion, the introduction is not very cohesive. The story starts with self-supervised learning, but this is not mentioned in the abstract, and it feels a bit detached from the rest. I would personally recommend the use of a whole introduction section (without subsections) and tailored to reflect the message flow from the abstract. Personal note, I like when the application setting is mentioned at the beginning of the introduction as a motivation, especially in papers like this where the biological application makes up the entirety of the experiments.

3. **Baselines:** One aspect I did not understand is why many MIL/VAE baselines were not considered, as today they represent pretty popular methods for group representations and predictive tasks [1,2,3]. I think they should at least be mentioned. As of now, these methods are quite popular for this task. I think the paper would significantly benefit from an Appendix section explaining the model setup, parameter selection, and all the related choices. In its current version, a lot of aspects are left underexplained.

4. **Vector-type of proportions.** I am not sure about this baseline; it feels very incomparable to the rest of the models, whose goal is to learn a cell-state representation. In my opinion, it remains unjustified how such a baseline was chosen over running established MIL models or pooling strategies on foundation model embeddings.

5. **Performance.** I don't think that the performance gain (especially in the first dataset) justifies the choice of scJEPA over e.g. DINO or pooled VAE. I also feel that a better understanding of the performance would be enabled by the use of error bars. As of now, performance increases of DINOv2 over scJEPA don't seem very marginal to me (e.g., 96.8% vs 93.6%).

6. **Results presentation.** In lines 256-257, it is mentioned that the embedding visualization for DINOv2 is also presented in Fig. 1. However, this does not happen. Moreover, I find Fig. 1 a bit hard to interpret. Is the coloring by patient ID? Why is one model better than another? The clustering seems pretty much patient-based in both settings.

7. **Stratification experiment.** I am not very sure about the significance of this analysis. I do not think that simply having a higher coefficient of variation is informative for capturing biological patterns. It is also not clear how the matrix is derived. Did you average the embedding of all cells perturbed by a certain cytokine per donor and simply compute a cross-Euclidean distance? In the ParseBio dataset, not all cytokines have a significant effect. Hence, I think a more reasonable analysis would be to assess whether the model can recognize a significant immune response. Also, I find it a bit strange that the CV's the authors hinted at were not reported. Is the evidence presented in the same experiment about cytokine co-clustering patterns supported by biological information? (Like differential gene expression).

8. **Drug response prediction.** This experiment is a bit unclear to me. What is the structure of the MLP? I think that, in general, perturbed states are very predictable from unperturbed states, even using gene expression. So I am not sure what the model is saying about embedding quality.

[1] Litinetskaya, Anastasia, et al. "Multimodal weakly supervised learning to identify disease-specific changes in single-cell atlases." bioRxiv (2024): 2024-07.

[2] Engelmann, Jan P., et al. "Mixed models with multiple instance learning." arXiv preprint arXiv:2311.02455 (2023).

[3] De Donno, Carlo, et al. "Population-level integration of single-cell datasets enables multi-scale analysis across samples." Nature Methods 20.11 (2023): 1683-1692.

**Questions:**

1. Allowing 70% of the variation to permeate the pointer tokens sounds quite high. Isn't the model offering too much information on the target? I think that some ablation study on the amount of variance retained would make this choice more solid. A similar thing is valid for the $\gamma$ parameter. What value was set for it? How was it selected? I think this is quite important to understand how you balance diversity with predictive power.

2. Why should the ARI be low in the cytokines dataset when using patient/donor labels? Isn't one of the main aims of the model that of segregating patients by their biology?

---

### Official Review · Reviewer_ry6H · 2025-10-27

**Soundness:** 2
**Presentation:** 2
**Contribution:** 2
**Rating:** 4
**Confidence:** 4

**Summary:**

The authors introduce SC-JEPA, a joint embedding predictive architecture designed for learning representations of multivariate population data, focusing on single-cell transcriptomics.
SC-JEPA aims to capture the structure of data sets where each data point comprises a set of multivariate samples, revealing insights about individual systems through the composition of their constituent elements.
The study explores the effectiveness of SC-JEPA in biomedical domains, particularly in scaling machine learning to large single-cell transcriptomics datasets for patient representation.

**Strengths:**

- SC-JEPA demonstrates strong batch integration, effectively removing dataset-specific variation.

- The model showcases predictive capabilities for tasks such as drug response prediction in unseen cell lines.

- SC-JEPA proposes a novel method for learning representations of multivariate population data.

**Weaknesses:**

- In Line 149, the authors claim that the proposed SC-JEPA framework can be applied to any dataset. This statement is overly strong. Since the experiments are limited to the biomedical domain, there is no evidence supporting the framework’s generalizability to other types of data. Additional experiments on non-biomedical datasets are needed to validate this claim.

- In Table 1, multiple evaluation metrics are reported, but it is unclear which of them are considered the most important for assessing model performance. The authors should clarify which metrics best reflect the effectiveness of their approach.

- From Table 1 and Table 2, it can be observed that Recall, Precision, CT R², and MSE do not outperform the baseline models. The authors should provide further discussion or analysis to explain these results.

**Questions:**

- In Line 154, how is the dataset divided into two disjoint subsets? Why was the 2:8 ratio chosen for this split?

In Line 159, the pointer token $p_{iT}$ is not clearly introduced. What is its specific role in the model, and why is it necessary?

---

### Official Review · Reviewer_repc · 2025-10-27

**Soundness:** 2
**Presentation:** 1
**Contribution:** 2
**Rating:** 2
**Confidence:** 3

**Summary:**

The authors propose a new model, sc-JEPA: a joint-embedding predictive architecture for single-cell transcriptomics. This model processes “multivariate population data”, a data structure where each object is represented by a set of $n$ vectors of size $d$. For example, a patient represented by $n$ single-cell molecular profiles. The goal of sc-JEPA is to learn meaningful patient-level embedding, where desideratas include removing batch effects, allowing informative patient stratification, and carrying signal for predictive tasks. Experiments on various datasets evaluate these 3 desideratas in comparison with a DINO-style, PCA, VAE, and cell-type proportion baselines.

**Strengths:**

- The paper tackles an important and difficult problem: the unsupervised learning of useful patient representations for downstream tasks such as patient stratification, a cornerstone of precision medicine.
- The authors propose an original approach to the problem through the use of JEPA.
- One of the main challenges lies in evaluating the learned representations, which, unlike in other fields of AI, cannot be assessed simply by computing accuracy on a downstream benchmark. To address this, the authors propose multiple complementary evaluation strategies.

**Weaknesses:**

**My main concern is the clarity of the paper. Many aspects are unclear, making it difficult for me to assess the soundness of the experiments.** I can easily get the overall ideas, but the precise implementation details are often unclear. I included a set of questions in the dedicated section to give examples. Overall, I do not understand on what data the model was trained. The datasets used for evaluation are not presented, so it is not easy to figure out what’s inside, the dimensions, etc. Many architectural details are missing (a diagram presenting the architecture would be helpful). The baselines are very succinctly presented, I am not sure what the baselines compute exactly.

**Questions:**

Some questions related to the lack of clarity.
- On what data was the sc-JEPA model trained?

- The baselines are barely described (paragraph l.226).  How is a vector of cell-type proportions computed? What’s the dimension? How to compute PCA when each element of the data matrix is a vector (as stated l.105)? How is DINOv2 adapted to the data at hand?

- L.252, when discussing technical integration, the paper only mentions that a dataset was built by aggregating “all PBMC datasets from the cellxgene census together with the Sound Life cohort.” However, it is unclear what this dataset actually looks like—its dimensionality or structure remain unspecified. The same issue applies for other datasets. L. 296 notes that the dataset comprises blood samples from 12 patients, but does not specify more details.

- What architecture for the context and target encoders? For example, what is the embedding dimension?
- L.171, what kind of pooled summary representation?
- L.178: how does the predictor (a 4-layer Transformer) map concatenated context and pointer tokens to a prediction of the target embedding? The input and output sizes are different in my understanding.
- L.183: how are dataset tokens added to the context embeddings?
- L.193: What are the input and output size, as well as architecture, of the MLP that produces the pointer tokens?
- Acronyms are used before being defined (or are not defined), eg “CLR-transformed cell type proportions” l.269 or “PBMC samples” l.245.
- L.340, k-means clustering (with k = 2–20) How is k chosen?
- l.341: “We summarized discriminability by computing the AUC of silhouette scores across cluster numbers.” How is AUC computed here?

---

### Official Review · Reviewer_xkEG · 2025-10-30

**Soundness:** 2
**Presentation:** 2
**Contribution:** 2
**Rating:** 2
**Confidence:** 4

**Summary:**

This paper proposes a new method for learning population-level representations with a joint embedding method.

**Strengths:**

The task is emergent and the idea is interesting. The model performance also looks promising.

**Weaknesses:**

Although the authors introduce a new method for learning population-level representation, the method still needs to be improved, and several important details in the method and experiment are missing. I have the following questions for this paper:

1. It seems that we cannot find the workflow plot of the proposed method, and thus, it is hard for people to get an overview of the paper. I think the authors need to include them in the updated version.

2. The datasets used by the authors are small. I think to fairly evaluate the effect of learning population-level representations, we need to use Onek1k and ROSMAP-scale datasets to make reliable conclusions.

3. Most of the tasks are not patient-specific and can be performed with cellular information. For example, we can also evaluate batch effect correction at the cell level, so what is the difference? Moreover, the metrics used for evaluating the two factors for correcting the batch effect are also missing. I suggest the authors refer to scVI and include the full evaluation settings to make a fair comparison.

4. There are also many methods that claim they are working on patient-level representations or also generate representations, for example, mcBERT, MrVI, and PaSCient, are also baselines for this task.

5. The authors should also include single-cell Foundation Models, such as scGPT, scFoundation, etc., in the evaluation to support their conclusion.

6. Figure 1 and other figures/tables are not clear. What is the meaning of the labels in Figure 1?

7. The authors should not use the 9th page to fill in information other than llm claim or limitations. I think they should reduce the content in the discussion part.

**Questions:**

Please check the weaknesses.

---

### Meta-Review · Area_Chair_fSRv · 2026-01-05

**Summary:**

This paper introduces sc-JEPA, a joint embedding predictive architecture for learning patient-level (population-level) representations from single-cell transcriptomics data. In this setting, each patient is modeled as a set of cellular profiles, and the goal is to learn embeddings that capture biologically meaningful variation while reducing technical confounding such as batch effects. Building on the JEPA framework, sc-JEPA predicts representations of target cell subsets from a context embedding derived from other cells within the same population, enabling self-supervised learning without explicit labels. The authors evaluate the learned representations across several desiderata, including batch effect removal, biological signal preservation, patient stratification, and predictive performance on downstream tasks such as perturbation or drug response prediction. Experiments on multiple single-cell datasets compare sc-JEPA against baselines including PCA, VAEs, DINO-style models, and cell-type proportion features.

**Reviewer Concerns:**

Reviewer xkEG is concerned that key methodological and experimental details are missing, the datasets are too small, important baselines and evaluations are absent, and the figures and presentation are unclear.
Reviewer repc finds the paper insufficiently clear, with missing descriptions of datasets, architectures, baselines, and training procedures, making it difficult to assess the soundness of the work.
Reviewer ry6H argues that the generalization claims are too strong given the limited biomedical evaluation, the reported metrics and results are unclear or underperform baselines, and important model components are insufficiently explained.
Reviewer JKCY believes the paper lacks strong methodological novelty, relies on unconvincing experimental evidence, omits key baselines, and presents results and analyses that are difficult to interpret or justify.

**Reviewer Scores:**

The authors did not provide a rebuttal to the comments made by the reviewers, so the reviewers may not change their score.

---

### Decision · Program_Chairs · 2026-01-26

Reject